# The Utilization of an Aloe Vera Rind By-Product: Deep Eutectic Solvents as Eco-Friendly and Recyclable Extraction Media of Polyphenolic Compounds

**DOI:** 10.3390/antiox13020162

**Published:** 2024-01-26

**Authors:** Georgia D. Ioannou, Katerina A. Ioannou, Atalanti Christou, Ioannis J. Stavrou, Constantina P. Kapnissi-Christodoulou

**Affiliations:** 1Department of Chemistry, University of Cyprus, 1678 Nicosia, Cyprus; gioann02@ucy.ac.cy (G.D.I.); ioannou.katerina@ucy.ac.cy (K.A.I.); achris54@ucy.ac.cy (A.C.); 2Department of Life Sciences, European University Cyprus, 2404 Nicosia, Cyprus; io.stavrou@euc.ac.cy

**Keywords:** *Aloe Barbadensis* Miller, aloe vera rind, ultrasound-assisted extraction, response surface methodology (RSM), deep eutectic solvents (DESs), green metrics, by-product, agrowaste, polyphenols, antioxidants

## Abstract

In this study, an optimized environmentally friendly procedure was employed to enhance the sustainable utilization of phenolic antioxidants derived from aloe vera rind by-products. The procedure involved the application of ultrasound-assisted extraction (UAE) in combination with deep eutectic solvents (DESs). Eleven different DESs and three conventional solvents were employed as extraction media for polyphenolic compounds. Choline chloride–citric acid (ChCl-CA) was selected as the most suitable extractant, considering its extraction efficiency in relation to the total phenolic content. The operating conditions of UAE were optimized and modeled by the use of response surface methodology in order to maximize the yield of total phenolics and antioxidant capacity. The optimal operational parameters for the UAE procedure were determined to be 16.5 min, 74% (*v*/*v*) DES in water, and a solvent-to-solid ratio equal to 192. HPLC analysis, which was performed on the optimum extract, revealed significant levels of phenolics present in the aloe rind. Efficient recovery of the extracted antioxidants was obtained by the use of solid-phase extraction (SPE) and polyamide cartridges. The ChCl-CA DES exhibited excellent recycling capability with a yield of over 90% through SPE. Finally, the greenness of the method was evaluated using the green AGREE and AGREEprep metrics. The results highlighted the sustainability and the greenness of the proposed extraction procedure for the aloe by-product.

## 1. Introduction

The aloe vera plant, scientifically known as *Aloe Barbadensis* Miller, is widely known for its remarkable pharmacological and therapeutic properties. Some of its beneficial properties include wound healing, anti-inflammatory and antibacterial effects, and laxative and antioxidant activities [1,2,3,4]. All these have led to the widespread utilization of the aloe plant and particularly its gel, in various industries, such as pharmaceuticals, food, and cosmetics [5]. Statistical analysis of economic data reveals a consistent upward trend in the revenue generated by the aloe industry [5].

Aloe vera belongs to the Liliacceae family, particularly the Aloe genus, which contains more than 400 species [6]. The aloe plant is a perennial succulent xerophyte with turgid pointed green leaves joined at the stem in a rosette pattern [7]. The leaf of aloe consists of three different layers: the inner gel, the latex, and the outer green rind. Aloe gel is the most extensively studied part of the plant due to its therapeutic properties, which are attributed to bioactive compounds. Aloe latex, a yellow liquid, is rich in anthrones, which are responsible for the laxative effect of the plant. The outer protective layer of the plant is a green thick rind, which is considered agrowaste and a by-product in aloe-related industries. The aloe rind corresponds to a significant percentage of the aloe leaves’ total weight, ranging from 20 to 30% [8].

In recent decades, the scientific community has aimed to increase the valorization of agricultural by-products and minimize waste generation in productive processes due to environmental issues [9,10]. A number of studies have recently reported the recovery of valuable bioactive compounds from agrowastes as renewable plant materials [11,12,13]. A significant example involves the aloe rind, which was found to contain significant amounts of polyphenolic compounds [14,15].

Polyphenols are a widespread class of compounds, formed through the secondary metabolism of plants. They are divided into subcategories, based on the number of phenol rings they contain and the structural differences in the binding between these rings [16,17]. The main classes of polyphenols are phenolic acids, flavonoids, stilbenes, and tannins. Phenolics, which are acknowledged for their role as natural antioxidants, have been extensively associated with a wide range of valuable effects on human health. In particular, polyphenols can protect important cellular components from oxidative damage through their effective scavenging of free radicals. As a consequence, the risk of degenerative diseases associated with oxidative stress is significantly reduced [18,19].

The extraction optimization of antioxidant compounds from aloe rind by-products has received limited attention, with only a small number of studies published on the subject. The antioxidant capacity of various parts of the plant (gel, rind, flower, and root) has been studied and compared, but the optimization of phenolic extraction has not yet been addressed [14,20,21]. Some studies have employed spectrophotometric assays to evaluate the antioxidant activity of aloe rinds, and their findings have indicated the high content of polyphenols [22,23]. Añibarro-Ortega et al. optimized the extraction procedure of a specific chromone, aloesin, from aloe rinds using alternative green solvents (glycerol and propylene glycol) along with a thermostatic magnetic stirring bath [24]. In another study, Solaberrieta et al. reported the optimization of the extraction procedure of polyphenols using non-conventional microwave-assisted extraction (MAE) [15].

The utilization of alternative extraction techniques results in an environmentally friendly sample preparation process due to reduced processing time and solvent consumption. Different non-conventional approaches are employed to extract phenolic components from plant matrices, including supercritical fluid extraction (SFE), pressurized liquid extraction (PLE), MAE, and ultrasound-assisted extraction (UAE) [25,26]. The application of UAE has acquired significant attention due to its numerous advantages. It is considered a simple, cost-effective, environmentally friendly, and highly efficient approach for phenolic extraction [12]. The enhancement of process efficiency by sonication is mainly attributed to the phenomenon of acoustic cavitation, leading to the breakdown of the cellular structure of plant material and the release of valuable polyphenolic compounds [27]. Additionally, this technique enables an efficient and selective extraction in short processing times, without the requirement of a high temperature. Consequently, it is considered to be more suitable for the recovery of thermally sensitive antioxidant compounds [28]. To the best of our knowledge, no prior research has, so far, been conducted to optimize the phenolics extraction procedure using UAE from aloe rinds.

The selection of the appropriate solvent is a critical step in sample preparation. In recent decades, there has been a shift toward sustainable and eco-friendly neoteric solvents as extraction media [29]. Among these solvents, ionic liquids (ILs) and deep eutectic solvents (DESs), classified as neoteric, have gained significant attention, with DESs exhibiting several advantages, including lower cost and an easier preparation process compared to ILs [30]. DESs consist of a hydrogen bond donor (HBD) and a hydrogen bond acceptor (HBA) in solid or liquid form [31,32]. The initial components can be varied, depending on the desired properties of the DES. Among them, the main categories include organic acids, monosaccharides, polyols, and amino acids.

DESs possess a range of notable properties, including low melting points, negligible vapor pressure, thermal stability, non-flammability, non-toxicity, and biodegradability [33,34]. These characteristics make DESs highly versatile and applicable in a wide range of fields, including catalysis and electrochemistry, and they can be used as additives in liquid chromatography and capillary electrophoresis and for extraction solvents [29,35,36,37]. In the existing literature, DESs have been reported to be employed as extraction media by the use of different extraction methods, mainly UAE and MAE, for the recovery of valuable components from plant matrices. According to these studies, DESs are more selective and efficient extractants of phytochemicals than conventional solvents [38,39,40]. However, to date, DESs have not been utilized for the recovery of high-value-added compounds from aloe vera.

Within this framework, different DESs were prepared and evaluated to determine the most suitable for the recovery of antioxidants from aloe rind by-products. The operational parameters from the UAE of aloe rind phenolics were optimized by spectrophotometric assays and modeled by response surface methodology (RSM). Phytochemical analysis of the extract under optimal conditions was performed by HPLC-PDA. Solid-phase extraction was applied on the aloe rind extract in order to recover the phenolics and the DES in different fractions. In addition, the greenness of the proposed extraction approach was evaluated using green metrics.

## 2. Materials and Methods

### 2.1. Chemicals and Reagents

The starting materials of DESs, choline chloride (≥98%), urea, ethylene glycol, 1,3-butanedion (99.5%), glycerol, citric acid monohydrate, D-fructose (≥99%), and D-sorbitol (≥98%) were purchased from Sigma-Aldrich (St. Louis, MO, USA). D-glucose anhydrous and citric acid monohydrate were acquired from CARLO ERBA Reagents SAS (Val de Reuil Cedex, France) and Merck (Darmstadt, Germany), respectively.

For HPLC analysis, acetonitrile (ACN) and methanol (MeOH) were provided by Sigma Aldrich (St. Louis, MO, USA), while trifluoroacetic acid (TFA) was purchased from Merck (Darmstadt, Germany). The analytical standards of catechin, quercitrin, and myricetin were obtained from HWI ANALYTIK GMBH (Rülzheim, Germany), while apigenin, epicatechin, quercetin, sinapic acid, caffeic acid, and gallic acid were purchased from Sigma Aldrich (St. Louis, MO, USA). Rutin was obtained from PhytoLab GmbH & Co (Vestenbergsgreuth, Germany).

The reagents for spectrophotometric assays, Folin–Ciocalteu reagent, sodium carbonate (Na_2_CO_3_), 2,4,6-tris(2-pyridyl)-s-triazine) (TPTZ), iron(III) chloride hexahydrate (FeCl_3_**·**6H_2_O), and sodium acetate were purchased from Sigma Aldrich (St. Louis, MO, USA), and DPPH was purchased from TCI (Tokyo, Japan).

### 2.2. Preparation of Deep Eutectic Solvents

In this study, the eleven DESs were prepared by the heating method. HBAs (choline chloride, glycerol, citric acid) and HBDs (organic acids, sugars, alcohols) were mixed, according to the appropriate molar ratio, as demonstrated in Table 1. The mixtures were then heated on a rotary evaporator at 80 °C until a clear liquid was formed. All DESs were stored in a desiccator at room temperature until the extraction procedure. The obtained DESs and their starting materials were analyzed by FTIR in order to confirm the formation of DESs (Appendix A).

### 2.3. Plant Material

*Aloe Barbadensis* Miller plants were generously provided by Puro Aloe Vera Care (Ormideia, Cyprus). Aloe vera leaves were cut from three-year-old plants during the summer and winter periods. Subsequently, aloe leaves were washed thoroughly with distilled water to remove any contaminants and soil particles, and with a knife, the green aloe skin was cautiously separated from the inner gel. The aloe rind was washed once again with distilled water in order to remove any residual gel. Then, the aloe skin was cut into small pieces and lyophilized (LyoDry Compact Benchtop Freeze Dryer, MECHATECH SYSTEMS LTD, Bristol, UK). The freeze-dried rind was ground to a fine powder using a Thermomix TM5 (VORWERK, Wuppertal, Germany) and was passed through a 250 μm sieve (Endecotts, London, UK) to obtain uniformly sized particles. Samples of different plants were homogenized, vacuum-packed, and stored at −20 °C until further extraction and analysis.

### 2.4. Ultrasound-Assisted Extraction of Polyphenols from Aloe Vera Rinds

The UAE of polyphenols was carried out by the use of a 500 W power and a 20 kHz frequency ultrasonic probe system (CY-500, Optic Ivymen System^®^, Barcelona, Spain). For the extraction, an appropriate amount of aloe rind powder was mixed with 10 mL of the extraction solvent, and the obtained mixture was exposed to ultrasound, following the experimental design conditions. In particular, the extraction procedure was performed using various extraction times, solvent-to-solid ratios, and concentrations of DESs. The ultrasound-treated solution was centrifuged in duplicate at 4400 rpm for 10 min, filtered, and stored at 4 °C until further analysis. All extracts were prepared in triplicate.

### 2.5. Determination of Total Phenolic Content

The TPC of aloe rind extracts was determined by the Folin–Ciocalteu spectrophotometric method, as described by Ozturk et al. [11], with minor modifications. Briefly, 80 μL of filtrated extract was diluted and mixed with 3.12 mL of distilled water. Then, a volume of 200 μL of Folin–Ciocalteu reagent 10% (*v*/*v*) in distilled water was added, and a vortex mixer was used for one minute. After 1–5 min, a 600 μL freshly prepared saturated solution of Na_2_CO_3_, 20% *w*/*v* was added into the reaction mixture and vigorously shaken. A UV-Vis spectrophotometer (UV-1900, Shimadzu, Tokyo, Japan) was used to measure the absorbance of the extracts (i) after 1 h of reaction in the dark and (ii) at room temperature at 750 nm. Gallic acid was employed as a reference standard, and the quantification of total phenolic content was expressed as the mg of gallic acid equivalents (GAE) per g of dried aloe rind. All analytical procedures were conducted in triplicate, and the outcomes were presented as the mean value along with its associated standard deviation.

### 2.6. Determination of Radical Scavenging Activity by DPPH

To evaluate the antioxidant capacity of the phenolic extracts by scavenging of free radicals of 2,2-diphenyl-1-picrylhydrazyl (DPPH), the method followed in this study was previously described by Xu and Chang [41]. An ethanolic solution of DPPH (0.1 mM) was freshly prepared. A volume of 3.8 mL DPPH solution was mixed with 0.2 mL of aloe rind extract and vigorously shaken. The mixtures were kept under light-obscured conditions at room temperature for a duration of 30 min, and the decrease in absorption was measured at 517 nm by use of a UV-Vis spectrophotometer (UV-1900, Shimadzu, Tokyo, Japan). The same amount of DPPH solution and a volume of 0.2 mL of the extraction solvent, without the sample, were added for the preparation of a blank sample, and its absorption was measured (t = 0 min). Radical scavenging activity (%) was calculated as [((1 − Abs_ext_)/Abs_bl_) × 100], where Abs_ext_ is the absorption of extract solution and Abs_bl_ is the absorption of a blank sample. Trolox was employed as a reference standard, and the outcomes of antioxidant activity based on DPPH were expressed as the mM of Trolox equivalents (TE) per g of dried aloe rind.

### 2.7. Determination of Ferric Reducing Antioxidant Power (FRAP)

The antioxidant activities of aloe rind extracts were evaluated through a reduction in ferric ion (Fe^3+^) to ferrous ion (Fe^2+^), according to the FRAP assay method, as outlined by Benzie and Strain [42]. Firstly, the working solution was prepared by adding 300 mM acetate buffer (pH 3.6), 10 mM TPTZ in 40 mM HCl, and 20 mM FeCl_3_. 6H_2_O solutions at a ratio of 10:1:1 were incubated at 37 °C for 30 min. Then, a 150 μL aliquot of the sample extract underwent a reaction with 2850 μL FRAP working solution in the dark at room temperature for a duration of 30 min. The absorbance of the blue-colored solution was measured at a wavelength of 593 nm by use of a UV-Vis spectrophotometer (UV-1900, Shimadzu, Tokyo, Japan). Results were expressed as μM TE per g of dried aloe rind using a Trolox as the standard compound.

### 2.8. Experimental Design and Statistical Analysis

The Box–Behnken design is one of the designs used to provide maximum information about a procedure using a minimum number of experimental data. Box–Behnken design was used to determine optimal conditions and evaluate the impact of three independent variables: extraction time (A), solvent-to-solid ratio (B), and percent content of the DES (C). The ranges of independent variables were evaluated in preliminary experiments, and Appendix A summarizes the three levels of independent factors that were investigated in the experimental design. The responses (dependent variables) were the TPC, DPPH, and FRAP in the dried aloe rind. A summary of the effect of extraction parameters on the responses of aloe rind extract is presented in Table 2. Design Expert version 13.0.5 software (Stat-Ease Inc., Minneapolis, MN, USA) was used for the analysis of variance (ANOVA) test in order to investigate the correlation between independent factors and responses.

### 2.9. Quantification of Polyphenolic Compounds through HPLC

For the identification and quantification of individual phenolic compounds in the aloe rind extracts, chromatographic analysis was performed by use of an HPLC system (Shimadzu, Tokyo, Japan) equipped with a photodiode array detector (PDA) (SPD-M20A). The chromatographic determination was conducted using the external standard method, employing ten analytical standards of phenolics (gallic acid, catechin, epicatechin, caffeic acid, rutin, sinapic acid, quercitrin, myricetin, quercetin, and apigenin). The chromatographic conditions employed in this study were similar to the ones used in the methodology described by Kumar et al. [43]. The PDA detector was set at 280 nm for the determination of all analytes under study. The analytical column (Venusil XSP C18—150 mm × 4.6 mm, 5 μm) was equipped with a pre-column composed of the same material, and the temperature was maintained constant at 25 °C. The mobile phase consisted of Milli Q water (mobile phase A) and 0.02% (*v*/*v*) TFA in ACN (mobile phase B), and the flow rate was 1 mL/min. A gradient elution program was implemented starting at 80% for the first five min. Mobile phase A was then decreased linearly to 60% at 8 min and subsequently to 50% at 12 min. Finally, mobile phase A increased gradually to 60% until 17 min and then was reset to its initial composition at 21 min, where it remained constant until the 25th min. The injection volume was 20 μL of extract for each analysis, and all experimental procedures were conducted in triplicate.

### 2.10. Recovery of Polyphenols and DESs through Solid-Phase Extraction

#### 2.10.1. Optimization of Polyphenol Recovery—Selection of SPE Cartridges

A solid-phase extraction (SPE) was performed in order to separate the polyphenols from the DES extracts. Three commercially available SPE cartridges were employed in this study, namely Maxi-Clean™ C18 (900 mg, GRACE, Tyrone, PA, USA), Oasis^®^ HLB (400 mg, Waters, Milford, MA, USA), and Discovery^®^ DPA-6S (500 mg, SUPELCO, Bellefonte, PA, USA). The evaluation criteria involved the recovery of polyphenols and the yield of the obtained DES. All SPE experiments were conducted using a vacuum manifold system with a 12-position rack (Visiprep™ SPE Vacuum Manifold, SUPELCO). The sorbent materials of the cartridges were first preconditioned with 5 mL of methanol and equilibrated with 5 mL of an acidified methanolic solution, 20:80 (*v*/*v*) MeOH:H_2_O acidified to pH 2.0 with HCl. Subsequently, 5 mL of an acidified standard mixture solution containing 30 μg/mL of each analyte and the appropriate amount of DESs in 20:80 (*v*/*v*) MeOH:H_2_O acidified to pH 2.0 with HCl were passed through each type of cartridge. The DES fractions were collected during this process. The acidification of the standard mixture solution was a necessary step to prevent the ionization of polyphenols and minimize their loss during the SPE procedure. The cartridges were washed with 5 mL of acidified water (pH 2.0, HCl). Finally, the fractions of analytes were obtained by eluting the bound phenolics from the cartridges. For C18 and HLB sorbent materials, elution was performed using 80:20 *v*/*v* MeOH:H_2_O (2 × 5 mL), while for polyamide DPA-6S cartridges, elution was carried out using a recommended mixture of acetone:H_2_O (80:20 *v*/*v*, 2 × 5 mL). The elution fractions were evaporated to dryness using a rotary evaporator and a lyophilizer. The collected DES fraction was weighed to determine its yield, and its FT-IR spectrum was acquired for comparison with the initial spectrum of the DES. The polyphenolic fraction was redissolved in 1 mL of MeOH, filtered through a 0.45 μm pore size membrane filter, and then injected into the HPLC system for further analysis.

#### 2.10.2. Recovery of Polyphenols and DESs from the Rind Extract

According to preliminary findings, the Discovery^®^ DPA-6S cartridge was employed for the recovery of polyphenols and DESs from aloe rind extracts. As mentioned, the polyamide cartridges were conditioned and equilibrated using 5 mL MeOH and 5 mL acidified methanolic solutions, respectively. The 15 mL sample mixture was prepared by three-fold dilution of the obtained DES extract in an acidified methanolic solution. It was then loaded into the sorbents, and the DES fraction was collected. The cartridges underwent a washing step that involved 5 mL of acidified water to eliminate any co-extracted compounds. Finally, the retained phenolics were eluted using a mixture of aqueous acetone (2 × 5 mL). The resulting fractions were subjected to evaporation until dryness was achieved. The FT-IR spectrum of the DES fraction was obtained and compared with the spectrum of the initial DES. The phenolics fraction was redissolved in 1 mL of MeOH, followed by filtration through a membrane filter with a pore size of 0.45 μm, and the resulting solution was introduced into the HPLC system for analysis.

## 3. Results and Discussion

### 3.1. Selection of a Suitable Green Solvent for Phenolic Extraction

The selection of an optimal solvent is a critical step in the extraction procedure since it affects the recovery of the targeted bioactive compounds from a given matrix. Different eco-friendly solvents, including three conventional (water, ethanol, and 50% ethanol) and eleven non-conventional (DESs) solvents, were examined through single-factor experiments in the medium level of independent variables (extraction time: 15 min, solvent-to-solid ratio: 150, and percent content of the DES: 70%). The selection of the appropriate solvent was based on the TPC of their extracts, as demonstrated in Figure 1.

The TPC values of the extracts of different types of solvents ranged between 9.89 and 52.75 mg GAE/g of dried rind. DESs demonstrated a greater TPC than conventional solvents, except for CA-Gly and Gly-Ur, whose values were at the same level as water and 50% ethanol. Higher TPC values were obtained when DESs that consisted of glycerol and citric acid, as well as the monosaccharides glucose and fructose as an HBD and choline chloride as an HBA were used. Although the two starting materials of CA-Gly combined with choline chloride demonstrated significant amounts of the TPC, the obtained value was considerably less than expected. Thus, it appears that HBDs and HBAs have a significant impact on the physicochemical properties of DESs. Consequently, these characteristics directly affect the ability of the DES to extract polyphenolic compounds from a plant matrix. The ChCl-CA demonstrated the highest TPC when it was utilized for the extraction of bioactive compounds from aloe vera rinds. Therefore, it was selected as the optimal solvent to conduct subsequent experiments.

### 3.2. Optimization of the Antioxidant Extraction Procedure from Aloe Rinds

RSM was applied through Box–Behnken design in order to determine the optimal operating parameters for the UAE of polyphenols from aloe vera rind by-products. It was employed to estimate the effect of multiple variables and their interactions on the responses with a minimum number of experiments. In particular, three independent variables, namely extraction time (A), solvent-to-solid ratio (B), and percent content of the DES (C) were investigated at three equally spaced levels (−1, 0, +1), which were selected based on preliminary experiments. The complete design included a total of sixteen experiments consisting of twelve combinations of the variables and four replicates performed at the central point. To mitigate the systematic error, the 16 experiments were performed in triplicate and random order. The derived extracts were subjected to analysis to determine the TPC and antioxidant capacity through DPPH and FRAP, which were defined as the dependent variables of the experimental design. The mean values of the responses obtained from three replicates are presented in Table 2. Under the established operating parameters, the experimental values varied between 36.6 and 69.7 mg GAE/g of dried aloe rinds, 7.4 to 17.1 mM TE/g of dried aloe rinds, and 799–2535 μM TE/g of dried aloe rinds for the TPC, DPPH, and FRAP, respectively.

#### 3.2.1. Statistical Analysis and Model Fitting

Individual statistical analyses were conducted for each dependent variable (TPC, DPPH, FRAP) within the experimental design. This approach was employed to determine the factors that exhibit a significant effect on the extraction process for each respective response. The outcomes from the statistical examination of variance (ANOVA) depicted in Table 3 indicate a strong level of significance for the extracted models (*p* < 0.0001) for all dependent variables. The R^2^ values serve as valuable metrics for evaluating the fitting of the model. The closer the R^2^ adjust value is to 1, the stronger the correlation of the experimental data to the mathematical equation [44]. The obtained R^2^ adjust values were 0.797, 0.984, and 0.775 for the TPC, DPPH, and FRAP, respectively, indicating good adequacy of the model to the response variables. The polynomial mathematical equations of response variables, which were obtained by fitting the experimental data, are presented below.
(1)YTPC=53.05−0.12 A+12.53 B−0.05 C
(2)$$Y_{DPPH}=11.75-0.15\;A+3.97\;B+0.62\;C-0.04\;AB+0.25\;AC+0.17\;BC+0.71\;A^{2}+0.19\;B^{2}-0.46\;C^{2}$$
(3)$$Y_{FRAP}=1785.51-112.00\;A+325.14\;B+449.65\;C$$

According to the regression analysis outcomes obtained using Design-Expert software, the best-fit model for the TPC and FRAP was the linear model, while the quadratic model was suggested for DPPH. The superior performance of the linear model for the TPC and FRAP can be attributed to the non-significant effect of quadratic and interaction terms on the respective responses. The statistical analysis performed confirmed the statistical significance of the coefficients derived from the polynomial models obtained. In the case of the TPC, only the linear term of the solvent-to-solid ratio (B) exhibited a significant effect. However, for FRAP, both the linear terms of the solvent-to-solid ratio (B) and the percentage of the DES (C) showed statistical significance. Notably, the percentage of the DES (C) made the most substantial contribution to the model, as indicated by its *p*-value, which was less than 0.0001. The variables with the largest impact on DPPH response were the linear coefficient of the solvent-to-solid ratio (B) and the percent content of the DES (C), along with the quadratic coefficient of extraction time (A^2^) and the percent content of the DES (C^2^). Conversely, the linear coefficient of extraction time (A) did not exhibit any notable effects on the TPC, DPPH, and FRAP values (*p* > 0.05). Based on all response variables, it can be concluded that the solvent-to-solid ratio is the most critical factor that affects the recovery of polyphenols from aloe by-products.

#### 3.2.2. Effect of Process Variables

As previously indicated, the solvent-to-solid ratio has been identified as the most important parameter in the extraction of these valuable bioactive compounds. Based on the experimental outcomes, an increase in the solvent-to-solid ratio leads to an elevation in the TPC and antioxidant activity (DPPH, FRAP) values. These findings confirmed that an increase in the amount of solvent promotes the interactions between the plant matrix and solvent, thereby enhancing the mass transfer of polyphenolic compounds and, subsequently, the extraction efficiency [11,45]. A lower ratio leads to the rapid saturation of the solvent, which, in turn, limits the quantity of extractable polyphenols [39].

The addition of water into the DES results in a reduction in the viscosity and an increase in the polarity of the extraction solvent. The presence of water in the DES affects the physicochemical properties of the DES, facilitating a more effective penetration of the solvent into the ruptured plant cell, enhancing the mass transfer and, consequently, improving the recovery of polyphenols [46]. On the other hand, excessive amounts of water (>50%) may negatively impact extraction efficiency by weakening or disrupting the intermolecular hydrogen bond interactions within the DES starting materials [38,46]. A water composition ranging from 20 to 40% was selected in order to avoid any interference with the DES. The composition of the DES and water in the extraction solvent significantly affects the antioxidant capacity of the extracts. In particular, an increase in the percentage of the DES leads to higher values of FRAP and DPPH.

The extraction time of bioactive compounds from a plant matrix plays a crucial role in the optimization of sample preparation. A prolonged extraction time may cause a degradation of sensitive polyphenolic compounds, while a reduction in the time may not result in the expected release and extraction of polyphenols from the plant cell [47,48,49]. Surprisingly, based on statistical analysis, the extraction time was found to have a non-significant effect on the recovery of valuable polyphenols from aloe rinds. This finding was unexpected, considering that similar studies performed on other plant matrices, such as ginger, pomegranate by-products, and cranberry, where UAE was performed with the DES as an extraction solvent, demonstrated a significant impact on extraction time [50,51,52]. Solaberrieta et al., who optimized the MAE of polyphenols from aloe rinds, reported that the extraction time and temperature were not important factors affecting antioxidant recovery and, therefore, the TPC, DPPH, and FRAP [15]. Given the consistency of our results with the above-mentioned finding, it is concluded that aloe vera rind is a unique plant matrix, and the sample processing time does not influence the release of phenolics.

#### 3.2.3. Optimal Conditions and Model Validation

Based on the acquired polynomial equations of antioxidant activity and the TPC, three-dimensional response surface plots were constructed to visually depict the correlation between the operating parameters and the response variables. Figure 2 exhibits these response surface plots, illustrating the association between the independent variables, namely the percent content of the DES and the solvent-to-solid ratio with the TPC, DPPH, and FRAP of aloe vera rinds.

In order to determine the optimum level of each operating parameter, a multiple response optimization was performed, with the objective of maximizing the TPC and the antioxidant activity of the extract. The results obtained indicated that the optimum conditions were as follows: an extraction time of 16.5 min, a solvent-to-solid ratio equal to 192, and an extraction solvent consisting of 74% (*v*/*v*) of the DES in water. The predictive ability of the response surface models was evaluated by conducting extractions under the estimated optimal conditions. The experimental results were in agreement with the predicted values, with a percentage of error ranging from 0.62 to 2.20% (Table 4). The strong correlation between the experimental and predicted values confirms the reliability and effectiveness of the models to correlate the independent and dependent variables and, consequently, to determine the optimal extraction conditions of phenolic components from aloe vera rinds.

The quantitative comparison of the results obtained from the analysis of aloe vera rinds, in terms of total phenolics and antioxidant activity, with the literature proved to be a challenging task. The main difficulty encountered was the variation in units of measurement used to express the results. In particular, FRAP values can be expressed in terms of either Fe^3+^ or gallic acid equivalents, while DPPH values can be described as % inhibition or IC_50_ [14,53,54]. For comparison purposes, the optimum DPPH value was expressed as 78.7 ± 0.62% inhibition. The DPPH results obtained in this study demonstrated a comparable antioxidant potential to that reported by Bushra et al. (77.6% inhibition), who employed conventional extraction techniques using 80% methanol as the extraction solvent [55]. However, the % inhibition of DPPH of aloe rind extracts obtained in the present work was higher than the values reported by Lopez et al. (58.8 ± 0.4%) and Hu et al. (39.7 ± 0.07%) [14,54]. In terms of the TPC, our findings are higher than those reported by Hossen et al. (3.52 mg GAE/g sample) [53]. Therefore, based on these comparisons, it can be concluded that the optimized operating conditions of the aloe rind extraction procedure employed in this study yielded higher levels of total phenolics and antioxidant capacity compared to the literature. Consequently, this optimization led to a greater recovery of valuable bioactive compounds.

### 3.3. Phenolic Profile of Aloe Rind Extracts—HPLC Analysis

Despite the fact that spectrophotometric assays enable rapid screening for the determination of polyphenolic compounds, they are susceptible to the potential overestimation of both phenolic content and antioxidant capacity due to cross-reactions with other reducing components [12,56]. Consequently, it became imperative to employ a chromatographic analysis for the identification and quantification of individual phenolics present in aloe rind extracts.

A rapid and accurate HPLC method was employed to determine the phenolic antioxidants. The method was validated prior to analysis in terms of precision, limit of detection, and limit of quantitation, as listed in Appendix A. All analytes exhibited excellent linearity, with the coefficients of determination varied above 0.997. The precision of the analytical method was evaluated by relative standard deviations (RSDs) of peak areas. The RSD values for repeatability (intraday precision) and reproducibility (interday precision) were lower than 1.04% and 2.68%, respectively. The limit of detection (LOD) and quantification (LOQ) of the method ranged from 0.1 to 0.9 μg/mL and 0.3 to 2.8 μg/mL, correspondingly.

The qualitative and quantitative results obtained from the chromatographic determination of aloe rind extracts, under the optimum extraction parameters, are presented in Table 5. The aloe rind was found to consist of a wide range of polyphenolic components. As demonstrated in Appendix A, the aloe rind extract is a complex plant matrix, with a variety of phytochemicals. Therefore, the detection of phenolics was performed by spiking the sample with pure individual standards. Among the ten phenolic compounds examined, the chromatographic method successfully detected eight. Myricetin exhibited the highest abundance among the antioxidants present in the aloe rind, surpassing catechin by approximately 70% in terms of concentration. Catechin, gallic acid, rutin, apigenin, quercetin, and epicatechin exhibited significant contributions to the overall phenolic content of the aloe rind, while caffeic acid was found to be present in relatively minor quantities compared to the aforementioned analytes.

In comparison with other studies, significant variations were noted in regard to the type and composition of phenolic compounds [14,15,57]. The variances observed in the phenolic profile are indicative of the impact of several factors, including geographical origin, plant variety, harvest season, soil characteristics, the extraction process, and the analytical method employed. For example, according to Lopez et al., catechin was found to be the most prevalent phenolic compound, followed by sinapic acid, while, according to our findings, sinapic acid could not be detected by HPLC in the optimum extract [14].

### 3.4. Recovery of Polyphenols and DESs from Aloe Rind Extracts

The isolation and recovery of extracted compounds from extracts is challenging due to the negligible vapor pressure of DESs [58,59,60]. In recent years, several approaches have been proposed to facilitate the recovery of valuable extracted compounds, including the applications of anti-solvents, macro-porous resins, and supercritical carbon dioxide [61,62,63,64,65,66]. The SPE procedure has proven to be a simple and efficient method for the isolation of phytochemicals from DES extracts [58,59,67]. Therefore, preliminary studies were performed to select the most suitable cartridge for recovering the extracted phenolics and the ChCl-CA, in different fractions, from the optimal aloe rind extracts. Three commercially available SPE sorbents (Maxi-Clean™ C18, Oasis^®^ HLB, Discovery^®^ DPA-6S), known for their favorable retention properties toward polyphenols, were evaluated for their ability to recover and isolate polyphenols and DESs. The recoveries of the polyphenolic compounds under study on different SPE cartridges, along with the standard deviations (SDs), are presented in Figure 3. Discovery^®^ DPA-6S sorbents demonstrated superior recovery ability of the target compounds and were consequently selected as the most suitable cartridges. A polyamide resin cartridge (DPA-6S) is typically employed for the absorption of polar compounds, particularly polyphenols, through hydrogen bonding between the hydroxyl groups of the compounds and the amide groups of the resin [12].

Following the optimization of the SPE method, the procedure was performed utilizing the aloe rind extracts under optimum conditions. The recovery of phenolics was determined to be at satisfactory levels, ranging from 54.3 to 96.1%. The lowest recovery values were observed for epicatechin and myricetin. The obtained DES fraction exhibited a yield of 95.98% ± 2.86. For confirmation purposes, FTIR experiments were conducted. The spectroscopic analysis confirmed a high degree of similarity between the findings and the initial DES (Appendix A). These findings highlight the ability of the DES to be recycled and employed in subsequent extraction cycles.

### 3.5. Green Evaluation Using Green Metrics

The green metrics have emerged as innovative tools in order to assess the greenness of analytical methods. The AGREE^®^ (version 0.5) [68] and AGREEprep^®^ (version 0.91) [69] tools are gaining attention nowadays due to their easy, user-friendly, and open-access software. Both metric tools employ a scoring system ranging from 0 to 1, along with a red-yellow-green scale, indicating the strengths and weaknesses of the analytical method. An overall score greater than 0.6 is considered indicative of a green method. The AGREE^®^ Calculator metric provides information about the entire analytical methodology. It consists of 12 criteria that align with the 12 categories defined by the Principles of Green Analytical Chemistry. Various parameters of the method, such as the nature and volume of reagents, energy consumption, generated waste, and the number of procedural steps, are taken into consideration. In contrast, the AGREEprep^®^ Calculator metric tool focuses on sample preparation and depends on 10 environmental impact criteria, including the type of solvents, sample size, waste generation, and energy consumption.

The present extraction procedure was compared with other methodologies described in the literature for the recovery of bioactive compounds in aloe vera rind samples, as illustrated in Table 6. Pictograms representing the chosen studies were generated using AGREE and AGREEprep software. The current sample preparation approach, which employs UAE combined with the DES, accomplished the greenest methodology, with an overall score greater than 0.7 in both metrics. The main drawback of the proposed procedure is the requirement to conduct sampling of the plant material (criteria 1). The substitution of organic and hazardous solvents with the DES contributes to the development of a more sustainable and green analytical method. In addition, the selection of UAE and the reduced process time contribute to low energy consumption. The HPLC-PDA system proved to be suitable for the chromatographic analysis of the DES extracts. However, the use of a mass spectrometer, as a detector in HPLC, may lead to potential damage due to the low volatility of the DES, the increased energy consumption, and equipment cost [70]. Despite these challenges, it is important to note that identifying complex mixtures with overlapping peaks in their chromatogram is still not possible without a mass spectrometer detector.

## 4. Conclusions

In the present study, the ultrasound-assisted extraction of phenolic antioxidants from aloe rind agrowaste, using ChCl-CA DES as the suitable green solvent, was optimized by RSM. Under the optimum operating conditions (16.5 min, solvent to solid 192, and 74% DES), high values of the TPC and antioxidant activity (DPPH and FRAP) were obtained. The aloe rind extract was subjected to chromatographic analysis, which revealed a notable presence of polyphenolic compounds, with myricetin identified as the most abundant compound. Furthermore, in order to recover the bioactive components and DESs from the extract, SPE utilizing DPA-6S cartridges was employed. This method exhibited excellent recovery of valuable phenolics, thereby demonstrating the recyclability of the DES. The greenness of the method was evaluated by employing green metrics. The extraction procedure achieved a score greater than 0.7, indicating that it is aligned with the concept of green chemistry. In summary, the combination of UAE and the DES provides a sustainable, efficient, and alternative methodology for the recovery of polyphenols from aloe vera rinds. This integrated approach is promising for the widespread application and utilization of aloe vera by-products on an industrial scale.

## Figures and Tables

**Figure 1 antioxidants-13-00162-f001:**
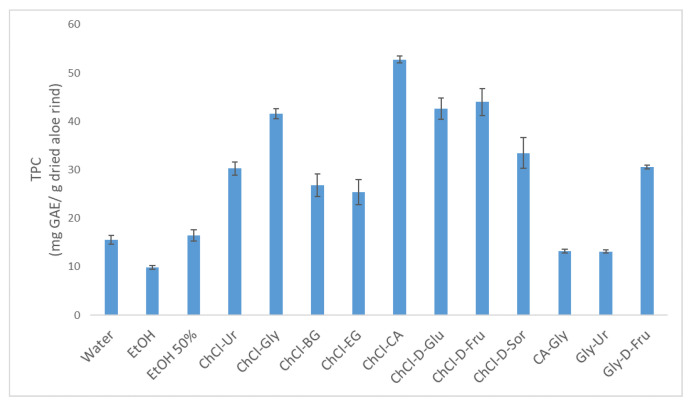
The TPC of aloe rind extracts using different green solvents. Extraction conditions: extraction time, 15 min; solvent-to-solid ratio, 150; DES percentage, 70%. Error bars indicate SD (n = 3).

**Figure 2 antioxidants-13-00162-f002:**
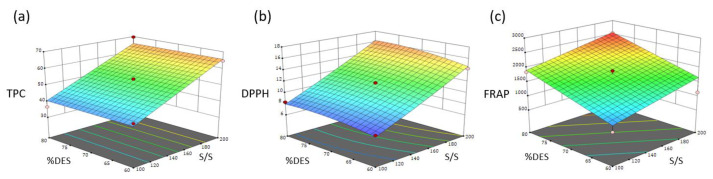
Three-dimensional response surface plots demonstrating the effect of the solvent-to-solid ratio and percentage content of the DES on (**a**) the TPC, (**b**) DPPH, and (**c**) FRAP.

**Figure 3 antioxidants-13-00162-f003:**
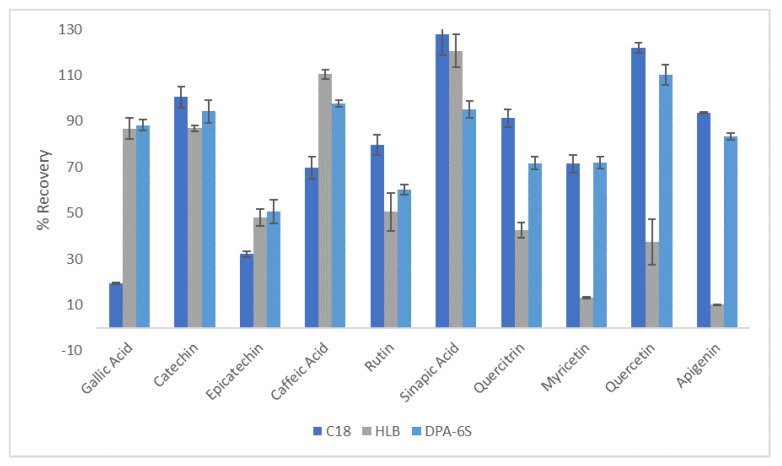
Recovery of the polyphenolic compounds on different SPE cartridges.

**Table 1 antioxidants-13-00162-t001:** Composition and abbreviation of the prepared DESs.

HΒA	HBD	Abbreviation	Molar Ratio
Choline Chloride	Ethylene Glycol	ChCl-EG	1:2
	Glycerol	ChCl-Gly	1:1
	Urea	ChCl-Ur	1:2
	1,3-Butylene Glycol	ChCl-BG	1:2
	Citric Acid	ChCl-CA	1:1
	D-Glucose	ChCl-D-Glu	2:1
	D-Fructose	ChCl-D-Fru	2:1
	D-Sorbitol	ChCl-D-Sor	1:1
Citric Acid	Glycerol	CA-Gly	1:2
Glycerol	Urea	Gly-Ur	1:1
	D-Fructose	Gly-D-Fru	3:1

**Table 2 antioxidants-13-00162-t002:** Box–Behnken design of the UAE of independent factors and experimental results for the TPC (mg GAE/g of dried aloe rinds), DPPH (mM TE/g of dried aloe rinds), and FRAP (μM TE/g of dried aloe rinds).

Run	Independent Factors			Responses		
	Time (min)	Solvent/Solid (mL/g)	% DES	TPC	DPPH	FRAP
1	15	100	60	42.1	7.4	799
2	10	150	60	45.8	11.9	1570
3	20	150	60	52.1	11.0	1423
4	15	200	60	64.6	14.3	1132
5	20	100	70	40.5	8.3	1338
6	10	100	70	49.7	8.4	1655
7	15	150	70	53.8	11.8	1889
8	15	150	70	53.9	11.8	1889
9	15	150	70	53.8	11.7	1889
10	15	150	70	53.8	11.8	1889
11	20	200	70	65.4	16.8	2234
12	10	200	70	69.5	17.1	2339
13	15	100	80	36.6	8.3	1846
14	10	150	80	45.9	12.6	2234
15	20	150	80	52.0	12.6	1907
16	15	200	80	69.7	16.0	2535

**Table 3 antioxidants-13-00162-t003:** ANOVA results for the TPC, DPPH, and FRAP models of aloe rind extracts.

Source	Sum of Squares (SS)	Degree of Freedom (DF)	Mean Square (MS)	F-Value	*p*-Value
TPC model				20.59	<0.0001 ***
A	0.1	1	0.1	0.01	0.9437
B	1256.5	1	1256.5	61.77	<0.0001 ***
C	0.1	1	0.1	0.01	0.9780
Residuals	244.1	12	20.34		
**R^2^ = 0.8373, R^2^_adj_ = 0.797, R^2^_pred_ = 0.757**					
DPPH model				103.69	<0.0001 ***
A	0.2	1	0.2	1.25	0.3070
B	125.9	1	125.9	886.43	<0.0001 ***
C	3.1	1	3.1	21.76	0.0034 **
AB	0.0	1	0.0	0.03	0.8588
AC	0.2	1	0.2	1.73	0.2370
BC	0.1	1	0.1	0.86	0.3908
A^2^	2.1	1	2.1	14.33	0.0091 **
B^2^	0.1	1	0.1	0.98	0.3602
C^2^	0.8	1	0.8	5.85	0.0497 *
Residuals	0.9	6	0.1		
**R^2^ = 0.994, R^2^_adj_ = 0.984, R^2^_pred_ = 0.899**					
FRAP model				18.26	<0.0001 ***
A	1.0 × 10^5^	1	1.0 × 10^5^	2.14	0.1688
B	8.5 × 10^5^	1	8.5 × 10^5^	18.07	0.0011 **
C	1.6 × 10^5^	1	1.6 × 10^5^	34.56	<0.0001 ***
Residuals	5.6 × 10^5^	12	46,798.8		
**R^2^ = 0.8203, R^2^_adj_ = 0.775, R^2^_pred_ = 0.733**					

* Significant at *p* ≤ 0.05. ** Significant at *p* ≤ 0.01.*** Significant at *p* ≤ 0.001.

**Table 4 antioxidants-13-00162-t004:** Experimental and predicted values in the optimum extraction conditions.

Dependent Variables	Predicted Value	Experimental Value	Percentage Error (%)
TPC (mg GAE/g)	63.56	64.96 ± 3.04	2.20
DPPH (mM TE/g)	15.51	15.69 ± 0.35	1.19
FRAP (μM TE/g)	2203	2217 ± 39	0.62

**Table 5 antioxidants-13-00162-t005:** Quantitative data for the HPLC analysis of the optimum aloe rind extract.

Compound	Concentration (mg/g Dried Aloe Rind)	% Content
Gallic acid	1.37 ± 0.14	14.2
Catechin	1.44 ± 0.09	14.9
Epicatechin	0.91 ± 0.04	9.4
Caffeic acid	0.21 ± 0.02	2.2
Rutin	1.20 ± 0.04	12.4
Sinapic acid	ND	0
Quercitrin	ND	0
Myricetin	2.45 ± 0.04	25.4
Quercetin	0.98 ± 0.03	10.1
Apigenin	1.11 ± 0.04	11.4
Total	9.67	100

ND: non-detected.

**Table 6 antioxidants-13-00162-t006:** Comparison of the extraction procedure from aloe rinds with the developed method.

Extraction Method	Extraction Time	Extraction Solvent	Mass of Aloe Rind	Solvent Volume	Detection System	AGREE	AGREEprep	Ref.
Soxhlet	12 h (4 × 3 h)	Hex, Ace, EtOH, MeOH	10 g	700 mL (4 × 175 mL)	HPLC-DAD	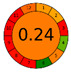	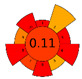	[14]
MAE	36.6 min	80% EtOH	1.5 g	50 mL	HPLC-ESI-MS/MS	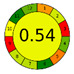	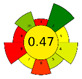	[15]
Stirring	3 days	Acidic MeOH	100 g	500 mL	UV	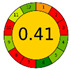	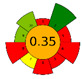	[53]
UAE	16.5 min	74% DES (ChCl-CA)	0.052 g	10 mL	HPLC-PDA	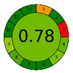	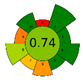	This work

## Data Availability

Data are contained within the article and Appendix A.

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
