# Peer review of "The Utilization of an Aloe Vera Rind By-Product: Deep Eutectic Solvents as Eco-Friendly and Recyclable Extraction Media of Polyphenolic Compounds"

_antioxidants, 2024, doi:10.3390/antiox13020162_

Round 1

Reviewer 1 Report

Comments and Suggestions for Authors

The authors gave good example of optimizing the extraction of phenolic compounds from plant materials. It is especially necessary to note the use of deep eutectic solvents (DES), which are the promising class of ionic liquids, as well as the AGREE and AGREEprep algorithms, which allow one to estimate greenness of the extraction methods.

There are the following minor remarks:

- The precision of the HPLC determination of individual phenolic components appears to have been determined using pure individual standards. The excellent metrological data obtained are shown in Table S2. However, a real chromatogram represents a superposition of overlapping peaks (Figure S12). In this case, the accuracy of determining individual components may be noticeably lower. The authors should discuss this point.

- The experimental values of TPC and DPPH are given in the article with an accuracy of 3-4 significant digits, but the values of FRAP are given with an accuracy of 5-6 significant figures (Tables 2,4). For FRAP values, it is quite enough to give 4 significant digits. For example, in Table 4 it should be 2217±39.

- The authors wrote: “However, the use of a mass spectrometer, as a detector in HPLC, may lead to a potential damage due to the low volatility of DES, the increased energy consumption and equipment cost [70]” (lines 515-517). It should be added that complex mixtures with overlapping peaks are still impossible to identify without mass spectrometry.  

Reviewer 2 Report

Comments and Suggestions for Authors

This research is very important and brings valuable information with immediate practical applications. The research is part of the current direction of increasing the valorization of agricultural by-products and minimizing the generation of waste in the production process due to environmental problems. As a result, it valorizes extraction optimization of antioxidant compounds from aloe rind by-product using ecofriendly solvents and methods.

The paper is written according to the requirements of the journal. The objectives that the authors proposed were successfully met.

Therefore, the Introduction provides sufficient data on the state of knowledge of this problem and supports the novelty and originality of the proposed research. An adequate experimental design and correctly applied and modern methodology were used. The results are well presented and statistically interpreted. The iconography (graphs, figures, tables) is appropriate and correct, explicitly presented. The discussions interpret the results taking into account the results obtained in this research and the discussions take into account several other data from the literature related to this topic. The authors have consulted numerous recent specialist references.

Starting from specifying the results and limitations of the research carried out and presented in this manuscript, the authors should mention future research directions that could be approached in perspective. These could be specified in the conclusions.

Reviewer 3 Report

Comments and Suggestions for Authors

Please see in the attach.
